# Investigating the Association between Diabetic Neuropathy and Vitamin D in Emirati Patients with Type 2 Diabetes Mellitus

**DOI:** 10.3390/cells12010198

**Published:** 2023-01-03

**Authors:** Tahra Al Ali, Alizeh Ashfaq, Narjes Saheb Sharif-Askari, Salah Abusnana, Bashair M. Mussa

**Affiliations:** 1College of Medicine, University of Sharjah, Sharjah P.O. Box 27272, United Arab Emirates; 2Sharjah Institute of Medical Research, University of Sharjah, Sharjah P.O. Box 27272, United Arab Emirates; 3Clinical Science Department, College of Medicine, University of Sharjah, Sharjah P.O. Box 27272, United Arab Emirates; 4Diabetes and Endocrinology Department, University Hospital Sharjah, Sharjah P.O. Box 27272, United Arab Emirates; 5Basic Medical Science Department, College of Medicine, University of Sharjah, Sharjah P.O. Box 27272, United Arab Emirates

**Keywords:** diabetic neuropathy, vitamin D deficiency, type 2 diabetes mellitus, Emirati population

## Abstract

(1) Background: Vitamin D deficiency is a common public health problem in the United Arab Emirates (UAE) and globally, and interestingly, improvements in diabetic neuropathy after taking Vitamin D supplementation for a short time have been reported. Despite living in a country that is sunny all year round, hypovitaminosis D, indicated by an obvious low serum vitamin D level, has been recurrently noted in the UAE, as well as in the surrounding Arabian Gulf countries. This problem is receiving much attention and attracting clinical and academic interest. Therefore, the main objective of the present study is to identify the association, if any, between vitamin D deficiency and the development of diabetic neuropathy in the UAE population with T2DM. (2) Methods: a total of 600 Emirati patients (male and female) with T2DM, aged between 20 and 80, were recruited from University Hospital Sharjah (UHS). The medical records of the patients were reviewed and analyzed. (3) Results: The results of the present study showed that among the 600 patients, 50% were affected with diabetic neuropathy. Vitamin D level in patients with neuropathy were estimated to be around 20 ng/mL (IQR 14–25), and vitamin D levels were significantly higher (33 ng/mL (IQR 20–42)) among patients without neuropathy, with *p* < 0.001. Another important finding was that patients without neuropathy had a better vitamin D status, with only 19% being deficient and 18% having insufficient vitamin D levels, compared to patients with neuropathy, where 39% were deficient (vitamin D < 20 ng/mL) and 44% had insufficient vitamin D levels (20–30 ng/mL). (4) Conclusion: The findings of the present study show that the prevalence of vitamin D deficiency (low serum 25-hydroxy vitamin D 25-OHD level) is significantly high in diabetic neuropathy in Emirati patients with T2DM.

## 1. Introduction

Diabetic neuropathy is one of the microvascular complications of diabetes, causing morbidity and mortality in patients with Type 2 Diabetes Mellitus (T2DM). It is a common complication of T2DM, and can be prevented by early detection and treatment. Nevertheless, when considering T2DM complications, current predictions estimate that, globally, T2DM will be the seventh leading cause of death by 2030 [1,2].

Complications associated with T2DM are rising among the population of the United Arab Emirates (UAE), leading to morbidity or mortality [3]. A recent study among the UAE population reported hyperlipidemia (84%) as being one of the major complications, followed by neuropathy, dyslipidemia, retinopathy, and nephropathy [3]. Peripheral neuropathy has become one of the major microvascular complications, affecting 34–35% of UAE population with diabetes [3,4].

The prevalence of T2DM in the UAE population is one of the highest in the world, and it is expected to increase to 21.4% by 2030 [2]. Previous reports have shown that the prevalence of T2DM among the Emirati (UAE citizen) adult population aged 20–70 years (18.7%) is the second highest in the world [1,2,5].

A study published in 2014 investigated the prevalence and identified the causes of peripheral neuropathy in patients with T2DM in UAE, showing that the prevalence of neuropathy in males was 37.0%, and 20.1% in females (*p* < 0.001) [4]. The conclusion of the same study showed that diabetic neuropathy is a common health problem in patients in the UAE [4].

The causes of T2DM remain unknown, as there are several mechanisms at play that can lead to the development of this disease. There are many factors that can lead to or predispose people to developing T2DM, including genetic factors and many environmental factors [6]. The latter include decreased physical activity, consumption of unhealthy food, and obesity. However, there is evidence suggesting that vitamin D deficiency might contribute to the development of T2DM [7]. This can be explained based on the numerous epidemiological studies that have demonstrated an inverse relationship between vitamin D levels and insulin resistance, hyperglycemia, obesity and prevalence of T2DM [8].

Vitamin D is a fat-soluble vitamin which has two forms: ergocalciferol (D2) and cholecalciferol (D3). About 20% of vitamin D comes from our diet, while the remaining 80% is provided by the skin [9]. Both Vitamin D2 and D3 bind to vitamin D-binding protein (VDBP) in the blood before being transported to the liver, where they are metabolized into their active form (calcidiol). Then, this is metabolized again in the kidney into its bioactive form calcitriol [9]. In the UAE, and throughout the Gulf region, vitamin D deficiency is highly prevalent due to people’s limited hours of exposure to the sunlight [8].

The mechanism of action of Vitamin D in T2DM is believed to be related not only to the regulation of plasma calcium levels, which regulates insulin synthesis and secretion, but also to direct action on pancreatic β-cell function [9].

Vitamin D influences β-cell insulin secretion through an increase in intracellular calcium concentration through voltage-dependent calcium channels which facilitate the conversion of proinsulin to insulin [10].

Vitamin D deficiency represents an ongoing debate in the literature [11]. Nevertheless, most experts agree that 25 (OH)D of < 20 ng/mL can be considered vitamin D deficiency, while a level of 21–29 ng/mL can be considered to be insufficient [12].

Several studies have suggested that there is an association between serum level of 25-hydroxy vitamin D (25-OHD) and glucose metabolism, glycemic control, and risk of developing diabetes [7]. Clinical studies have suggested that vitamin D deficiency is more common in people with T2DM, and interestingly, improvements in diabetic neuropathy have been reported after taking vitamin D supplementation for a short time [11]. It has been reported that vitamin D increases insulin secretion and sensitivity to insulin, and improves β-cell function while reducing insulin resistance [13].

In addition, Mousa et al. studied the anti-inflammatory effect of Vitamin D on inflammatory markers in patients with T2DM, and the results of the study showed the beneficial effect of Vitamin D supplementation on inflammation, with a potentially important effect on diabetes [14].

Pittas, A.G. et al., in their study, recognized that T2DM was associated with systemic inflammation, and was linked to insulin resistance primarily through elevated cytokines, which may play a role in β-cell dysfunction by activating β-cell apoptosis. In addition, vitamin D may improve insulin sensitivity and promote β-cell survival by directly controlling the generation and effects of cytokines [15].

Previous studies have suggested that vitamin D deficiency is more common in people with T2DM, contributing to its pathogenesis in many ways, including impairment of insulin secretion from pancreatic β-cells, and it plays an important role in the pathogenesis of diabetic neuropathy [16]. Other prospective clinical studies have shown improvements in neuropathy with vitamin D supplementation [11,14]. Several studies have also suggested that vitamin D has an anti-inflammatory effect; however, the effect of vitamin D supplementation on inflammation has not yet been established [14,16]. One of the hallmarks of T2DM is low-grade inflammation, which can be caused by circulating cytokines. Increasing amounts of these circulating cytokines contribute significantly to insulin resistance in muscle and adipose tissue [7].

A number of studies, including clinical studies and observational studies (systematic reviews and meta-analyses), have suggested that vitamin D deficiency is more common in people with T2DM and it contributes to its pathogenesis in a significant manner [11,14,16]. Therefore, the main objective of the present study is to investigate the association, if any, between vitamin D deficiency and diabetic neuropathy in the UAE population with T2DM.

## 2. Materials and Methods

### 2.1. Methods

#### Study Population and Design

The current study is a retrospective, cross-sectional study that was conducted using patient medical records at the University Hospital Sharjah (UHS). The study obtained ethical approval from the Ethics committee of UHS (UHS-HERC-007-10032019).

Six hundred Emirati patients with T2DM were recruited from the diabetes clinics; given that this is a secondary care unit, a significant number of the patients experienced complications, including diabetic neuropathy. Three hundred out of the six hundred experienced diabetic neuropathy, while three hundred were without diabetic neuropathy. The following criteria were used for the diagnosis of diabetic neuropathy: (i) patients were assessed for distal symmetric polyneuropathy; (ii) the assessment included a careful history and either temperature or pinprick sensation (small-fiber function) and vibration sensation using a 128 Hz tuning fork (large-fiber function); and (iii) patients underwent annual 10 g monofilament testing to assess for feet at risk of ulceration and amputation. Finally, electrophysiological testing or referral to a neurologist was rarely needed for screening, except in situations where the clinical features were atypical, the diagnosis was unclear, or a different etiology was suspected. Atypical features included motor neuropathy being greater than sensory neuropathy, rapid onset, or asymmetrical presentation.

The inclusion criteria were Emirati patients with T2DM aged between 20 and 80 years, with and without diabetic neuropathy.

Exclusion criteria were patients having Type 1 diabetes, and/or patients aged less than 20 or older than 80 years. The medical records of the selected subjects were reviewed, and the following details were documented: (i) demographic data; (ii) medical history; and (iii) diagnosis of T2DM and diabetic neuropathy.

The following variables were included in the investigation: age, gender, height, weight and body mass index (BMI), HbA1c (normal: 6.5%), diabetes duration, vitamin D status (deficient vs. non-deficient), C-reactive protein levels (reference range: 0.0–9.0 mg/L), lipid profile including total cholesterol (normal range: 0.0–5.2 mmol/L), triglycerides (normal range: 0.40–1.82 mmol/L), HDL (normal range: >1.04–1.55mmol/L) and LDL (normal range: <0.00–4.00 mmol/L). Serum creatinine (normal range: 62–115 umol/L), Urine creatinine (normal range: 8400–2,2000 umol/L) and urine microalbumin (normal range: 3–30 mg/L) were also included. Vitamin D supplementation, including Oligocalciferol, Ergocalciferol, Calciferol and Vitamin D3, were considered.

### 2.2. Statistical Analysis

Counts and percentages are used to represent categorical variables. Normally distributed continuous data are presented as mean ± standard deviation (SD), whereas continuous data with skewed distribution are presented as median and median with interquartile range (IQR) (PMID: 24464827). For skewed continuous data, the values were log-transformed and used in univariate (Vitamin D, Hb1AC, total cholesterol, triglycerides, HDL, LDL, serum creatinine, urine creatinine, and microalbuminuria) and multivariate regression (Microalbuminuria) analyses (PMID: 17533212, PMID: 34666549, PMID: 20065202). Model calibration, defined as the agreement between observed and predicted outcomes, was determined using the Hosmer and Lemeshow test of goodness of fit (PMID: 7055134).

In the univariate analysis, continuous data were analyzed using t-test, while the categorical data were compared using χ2test. Independent factors of diabetic neuropathy were identified through the development of a logistic regression model using the enter method, which was adjusted for age, gender, and any other variables that were significant in the univariate analysis at *p* < 0.05. The Statistical Package for Social Sciences (SPSS) version 24 (IBM Corp, New York, NY, USA) was used to carry out all analyses.

## 3. Results

### 3.1. Anthropometric, Clinical Characteristics and Vitamin D Supplement Consumption of the Study Population

A total of 600 T2DM patients were recruited from UHS, Sharjah, UAE. The clinical and anthropometric characteristics are presented in Table 1, and the data for all variables are presented as total number (*n*), percentage (%), and mean and standard deviation (± SD) for continuous variables. The study sample included 247 (41%) males and 253 (58%) females. The mean age of the patients was 62.85 ± 11.35 years, and had had diabetes for more than 10 years.

The mean BMI was 31.1 kg/m^2^ ± 6.0, and approximately 52% of the patients were obese, with a BMI > 30 kg/m^2^. Out of 600, 363 (60.5%) patients were deficient in vitamin D and 237 (40%) were not deficient. Moreover, consumption of vitamin D supplements was documented in 490 (81.75%) of the total population. Most of these patients (76.8%) had a HbA1c level > 6.5%, and the vitamin D levels of 50% of the patients was < 30 ng/mL. Out of 600, 94 patients had vitamin D levels of < 20 ng/mL, 103 had vitamin D < 30 ng/mL and 22 were markedly deficient in vitamin D. The CRP levels of 53% of the patients were normal, with 87% of these patients having normal creatinine levels. A total of 54% participants had normal microalbuminuria levels, and lipid profiles, including total cholesterol, HDL, LDL, and triglycerides, were normal for the majority of the patients. A total of 69% of the patients had normal total cholesterol levels, and 62% had normal triglyceride levels.

### 3.2. Comparison of Anthropometric, Clinical Characteristics and Vitamin D Supplement Consumption of Patients between Group 1 (Neuropathy) and Group 2 (No Neuropathy)

To further investigate the variables that might be involved in the development of diabetic neuropathy, a comparison was conducted between patients with and without neuropathy. Table 2 shows the basic anthropometric and clinical characteristics, along with vitamin D supplement consumption, of patients with (group 1) and without (group 2) neuropathy. Of the 600 total patients, 300 experienced neuropathy as a complication, while the other 300 were diagnosed with T2DM without neuropathy. Data for all variables are presented as total number (*n*), percentage (%), and mean and standard deviation (± SD) for continuous variables. Evaluation of the results showed that gender, age and HbA1c level did not differ between the two groups. The majority of patients with diabetic neuropathy had a long-standing diagnosis of diabetes, >10 years, were overweight or obese, and were deficient in vitamin D compared to patients without neuropathy. Patients with neuropathy had low vitamin D levels, with a mean vitamin D level of 21.37 ± 11.9 (ng/mL) compared to T2DM patients without neuropathy. However, the consumption of vitamin D was similar between the two groups. No significant differences were noted in the lipid profile (total cholesterol HDL, LDL, triglycerides), creatinine, microalbumin urine or creatinine urine between the two groups.

Comparing the patients with and without neuropathy, it can be observed that there is no significant difference in mean age, gender, and mean BMI between the two groups. A majority of the whole study population, approximately 84%, were either overweight or obese, while 89% of patients with neuropathy had a higher BMI, *p* = 0.006, as shown in Table 3. The laboratory data indicated that the median vitamin D level for the whole study population was 24 ng/mL (IQR 17–34). For patients with neuropathy, this was 20 ng/mL (IQR 14–25), and the vitamin D level was higher (33 ng/mL (IQR 20–42)) among patients without neuropathy, at *p* < 0.001, which is highly significant. Patients without neuropathy had a better vitamin D status, as only 19% were deficient, while 18% had insufficient vitamin D levels, whereas among patients with neuropathy, 39% were deficient (vitamin D < 20 ng/mL) and 44% had insufficient vitamin D levels (20–30 ng/mL), at *p* < 0.001, showing that the difference was highly significant. Vitamin D was significantly lower in patients with (21.37 ± 11.9 ng/mL) compared to without (33.41 ± 15.7 ng/mL) diabetic neuropathy (*p* ≤ 0.001). A significant difference was also noted in microalbumin urine levels between the two groups, where the interquartile range (IQR) for patients with neuropathy was slightly higher (4–61 vs. 4–35, *p* = 0.020). However, no significant differences were noted between the two groups for other laboratory values like HbA1c, lipid profile and creatinine.

### 3.3. Correlation of Diabetic Neuropathy with Different Variables

In addition, multivariate regression analysis was performed to investigate the association between diabetic neuropathy and different variables. Several variables were assessed, including age, gender, BMI, vitamin D level, and microalbuminuria. As shown in Table 4, only vitamin D < 20 ng/mL (adjusted OR 2.63, *p* < 0.001), Log10 Microalbuminuria (adjusted OR 1.40, *p* 0.010) and BMI > 25 (adjusted OR 1.93, *p* 0.024) were significant predictors of diabetic neuropathy. The multivariate model also demonstrated good calibration, with a Hosmer–Lemeshow statistic equal to 0.8.

### 3.4. Strength of Association between Different Factors and Diabetic Neuropathy

Different variables were investigated regarding their association in the development of diabetic neuropathy, and it was found that the chances of developing diabetic neuropathy in patients with vitamin D levels < 20 ng/mL were significantly high, with an adjusted OR of 2.63, at *p* < 0.001, which is highly significant. As shown in Figure 1, along with vitamin D level, BMI > 25 kg/m2 (adjusted OR 1.93, *p* = 0.024) and Log10 microalbuminuria (adjusted OR of 1.40 *p* = 0.010) were also significantly associated with diabetic neuropathy.

## 4. Discussion

The current literature indicates that circulating vitamin D (25-hydroxy vitamin D) could be involved in the development and progression of diabetic neuropathy. In addition, vitamin D deficiency/insufficiency is an under-recognized problem, and perhaps it is an important contributing factor to diabetic neuropathy [11]. This study is the first to investigate the association between vitamin D deficiency and diabetic neuropathy in the UAE population.

The present study showed that individuals with diabetic neuropathy had lower vitamin D levels compared to those without diabetic neuropathy. In agreement with these findings, the study of Abdelsadek et al. showed that the mean serum vitamin D levels in patients with diabetic neuropathy were lower than in patients without neuropathy; they also found that 87.6% of their patients with diabetic neuropathy had vitamin D deficiency, with vitamin D level of < 28.3ng/mL, which is much higher than in patients without diabetic neuropathy, where only 45% exhibited vitamin D deficiency [16].

In addition, very similar findings were reported by Abdelsadik et al., suggesting that the majority of patients with diabetic neuropathy had diabetes for more than 10 years, and diabetic patients generally had a higher BMI, of 31 ± 4 kg/m^2^, than the healthy participants [16]. Another significant difference between the two groups noted in this study was related to microalbumin urine levels, as the IQR for patients with neuropathy was higher (4–61 vs. 4–35, *p* = 0.020). However, no significant differences were noted between the two groups in other laboratory values including HbA1c, lipid profile and creatinine.

In accordance with our present research study, Smith et al., in their study on the risk factors leading to diabetic neuropathy in subjects with T2DM and with diabetes < 5 years, found that obesity is one of the risk factors that significantly increases the risk of diabetic neuropathy [17].

Furthermore, the comparison between patients with (group 1) and without (group 2) neuropathy in our study suggests that inadequate vitamin D (either vitamin D deficiency or insufficiency, or vitamin D levels less than 30 ng/mL) was significantly more prevalent (*n*%) among cases with neuropathy than among cases without neuropathy, at *p* < 0.001 (Table 3).

Consequently, this suggests that low vitamin D levels are more prevalent in diabetic patients with neuropathy than in diabetic patients without neuropathy. These results match those obtained by other researchers, including Alamdari et al. and Abdelsadek et al., who reported that more diabetic patients with low vitamin D levels had neuropathy than those with normal vitamin D levels [16,18].

The present study confirmed the relationship between vitamin D deficiency and diabetic neuropathy, which corroborates the findings of previous studies [16,18,19,20]. Previous studies also support the finding that diabetic neuropathy increases with vitamin D deficiency [21,22,23]. Pietschmann et al. and Isaia et al. reported an association in their studies between low vitamin D levels and the occurrence of diabetes and impaired glucose tolerance [24,25]. Clinical studies, including Alamdari et al., Bajaja, et al., and others, have reported a significant relationship between vitamin D deficiency and diabetic neuropathy. Furthermore, Putz et al. recommended vitamin D supplementation in patients with diabetic neuropathy [18,19,21].

Likewise, the study of Shehab et al. demonstrated a significant relationship between vitamin D deficiency and the severity of clinical manifestation in diabetic neuropathy (sensory and neurological deficits) [26].

However, Scragg et al., Suzuki et al. and Putz et al., in their studies on vitamin D level in patients with T2DM, stated that there were limited and conflicting data on vitamin D deficiency in patients with diabetic peripheral neuropathy (DPN), and the data were not statistically significantly different from those in normal subjects, which could potentially be attributed to smaller sample sizes, geographical locations, and the effects of weather [6,19,20].

Compared to the previous mentioned studies this current study is the first to highlighting significant association between the vitamin D deficiency and diabetic neuropathy in the UAE.

The study of Alamdari et al. found that the serum vitamin D level was inversely correlated with the intensity of the impairment of nerve conduction velocity, at *p* = 0.001 [18].

Moreover, Abdelsadek et al., in their discussion, commented on the study of Kheyamis, which demonstrated that the expression of vitamin D receptors (VDR) was increased in diabetic neuropathy patients, and this upregulation of VDR was associated with the severity of neuropathy and peroneal nerve conduction velocity, with no difference in VDR expression being observed between painless neuropathy and painful neuropathy [16].

Celikbilek et al. in their study examined the relationships between serum vitamin D, VDBP and VDR, and diabetic neuropathy, finding that patients with diabetic neuropathy had significantly lower levels of vitamin D than patients without neuropathy, while the values of VDBP and VDR were similar between the two groups (diabetic patients with and without neuropathy) [27].

Saadi et al. in their study aimed to investigate the efficacy of daily and monthly vitamin D supplements in lactating and nulliparous women, finding that high doses of vitamin D supplements of 200 IU/day—60000 IU/ month were effective for safely increasing serum 25 (OH) D. However, only a small number of women achieved levels > 50 mmol/L. The authors concluded that when sunlight exposure was limited, higher doses of daily vitamin D might be required to reach adequate vitamin D levels. Meanwhile, monthly dosing was found to be a safe and effective alternative to daily dosing [28].

The study of Xiaohua et al. aimed to evaluate the association among vitamin D status, inflammatory cytokines, and painful DPN. They found that the prevalence of severe vitamin D deficiency < 10 ng/mL was more common in the painful DPN group than in the painless DPN and non DPN groups, at *p* = 0.01. The investigator also found that patients with painful DPN had significantly higher concentrations of interleukin 6 (IL-6) (*p* ≤ 0.01) and Tumor necrosis factor (TNF-α) (*p* < 0.01) than the other two groups. It was concluded that severe vitamin D deficiency was an independent risk factor for painful DPN. They also concluded that severe vitamin D deficiency may play a role in the painful DPN pathogenesis through elevated IL-6 and TNF-α level [29].

Herder et al. in their study found that higher serum levels of CRP, IL-6, TNF-α, and other inflammatory cytokines were associated with a higher risk of DPN [30].

Smith et al. in their study examined the relationship between glucose control, lipid profile, blood pressure and obesity and peripheral neuropathy risk, finding that there was a significant relation between obesity and increased risk of peripheral neuropathy, with a risk ratio of 2.9 *p* > 0.02 for obesity [17].

Research by Abdelsadek et al. on the role of vitamin D in the pathophysiology of DNP reported that animal studies revealed a connection between vitamin D deficiency and low levels of nerve growth factors (neurotrophins), which are necessary for the development and survival of both sympathetic and sensory neurons. Decreased levels of neurotrophins are associated with an increase in the damage caused by toxins, including in the case of hyperglycemia, which is directly linked to vitamin D deficiency. The latter is also associated with impairment of nociceptor function, worsening of nerve damage and lowering of the pain threshold [16].

## 5. Limitations of the Study

One of the main limitations of this study was the number of the sites involved; a multicenter study would be better to provide a good representation of UAE population. Future studies will include more recruitment sites in order to have a larger sample size of UAE participants. Moreover, as per the study design, controls were recruited by convenience sampling, and thus were not representative of the general population and were prone to selection bias.

## 6. Conclusions

In conclusion, vitamin D deficiency is an independent risk factor for diabetic complications, including diabetic neuropathy. This study showed that vitamin D deficiency plays a significant role in development of diabetic neuropathy. Therefore, it is essential to make sure that the vitamin D levels in all T2DM patients are within the normal range and vitamin D intake (supplementation) is essential if vitamin D deficiency and/or insufficiency are present in this group of patients. In addition, along with vitamin D levels, BMI > 25 kg/m^2^ (adjusted OR 1.93, *p* = 0.024) was also significantly associated with diabetic neuropathy.

Another significant difference between the two groups (with and without diabetic neuropathy) was noted in this study with respect to microalbumin urine levels, with the IQR for patients with neuropathy being higher (4–61 vs. 4–35, *p* = 0.020). However, no significant differences between the two groups were noted for other laboratory values, including HbA1c, lipid profile and creatinine. Therefore, further analysis of these factors and their relation to diabetic neuropathy is required to validate the present findings.

## Figures and Tables

**Figure 1 cells-12-00198-f001:**
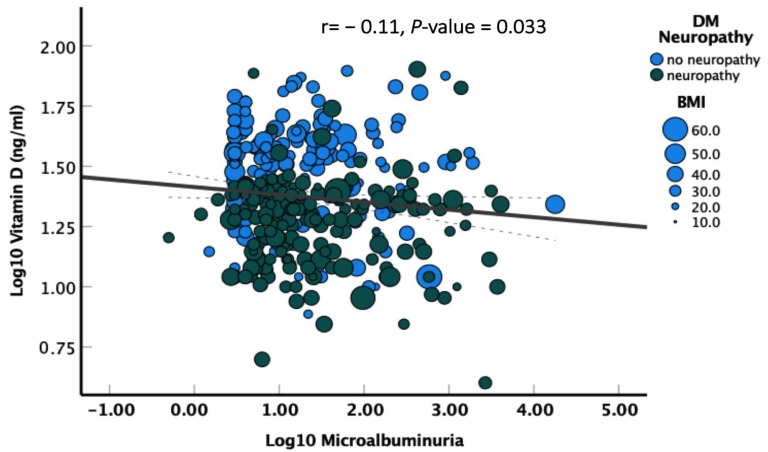
Correlation of log-transformed vitamin D and albuminuria values. Blue filled color circles represent DM with neuropathy, while green filled circles present DM without neuropathy. The size of the circles represent BMI values, with increasing values corresponding to larger circle size. Correlation was tested using Pearson’s test with significance indicated by *p*-values less than 0.05.

**Table 1 cells-12-00198-t001:** Basic anthropometric, clinical characteristics and Vitamin D supplement consumption of patients (*n* = 600).

Variables	*n*	%	Mean ± SD	Variables	*n*	%	Mean ± SD	
Gender				Vitamin D levels				
Females	353	59%		Low	300	50%	27.3 ± 15.2	
Males	247	41%		Normal	148	25%		
Age (years)				CRP				
20–40	36	6%	62.85 ± 11.35	High	206	34%	18.1 ± 33.1	
41–60	139	23%		Normal	315	53%		
61–80	425	71%		Total Cholesterol				
Duration of diabetes				High	107	18%	44.34 ± 1.15	
1–10 years	125	21%		Normal	416	69%		
11–20 years	120	20%		Triglycerides				
21–30 years	78	13%		High	133	22%	1.56 ± 0.82	
>30 years	34	6%		Normal	374	62%		
Height (cm)	159.3 ± 11.37			HDL				
				High	66	11%	1.48 ± 6.2	
weight (cm)	79.3 ± 16.8			Normal	270	45%		
BMI				Low	158	26%		
Normal weight	79	13%	31.1 ± 6.0	LDL				
Overweight	192	32%		High	62	10%	2.85 ± 3.26	
Obese	310	52%		Normal	462	77%		
Underweight	1	0%		Creatinine				
Vitamin D deficiency				High	69	12%	85.6 ± 52.8	
Yes	363	61%		Normal	522	87%		
No	237	40%		Microalbumin Urine				
Consumption of Vitamin D supplements				High	159	27%	176.8 ± 919.2	
Yes	490	82%		Normal	326	54%		
No	110	18%		Creatinine Urine				
Neuropathy				High	8	1%	7832.9 ± 5536	
Yes	300	50%		Normal	446	74%		
No	300	50%						
HbA1c								
<5.7%	24	4%	7.68 ± 1.65					
5.7–6.5%	101	17%						
>6.5%	461	77%						

Abbreviations: BMI, body mass index; HbA1c, glycated hemoglobin; CRP, C-reactive protein; HDL, high density lipoprotein; LDL, low density lipoprotein.

**Table 2 cells-12-00198-t002:** Comparison of anthropometric characteristics, clinical characteristics and Vitamin D supplement consumption of patients between group 1 (neuropathy) and group 2 (no neuropathy).

Variables	Neuropathy	No Neuropathy	Neuropathy	No Neuropathy	Variables	Neuropathy	No Neuropathy	Neuropathy	No Neuropathy
	*n* (%)	*n* (%)	Mean ± SD	Mean ± SD		*n* (%)	*n* (%)	Mean ± SD	Mean ± SD
Gender					Vitamin D levels				
Females (*n* = 353)	176 (49.8%)	177 (50.1%)			Low (*n* = 300)	209 (69.6%)	91 (30.3%)	21.37 ± 11.9	33.41 ± 15.7
Males (*n* = 247)	124 (50.2%)	123 (49.7%)			Normal (*n* = 148)	19 (12.8%)	129 (87.2%)		
Age (years)					CRP				
20-40 (*n* = 36)	7 (19.4%)	29 (80.5%)	63.45 ± 9.3	62.25 ± 13.0	High (*n* = 206)	123 (59.7%)	83 (40.2%)	19.5 ± 31.4	16.7 ± 34.9
41-60 (*n* = 139)	81 (58.2%)	58 (41.7%)			Normal (*n* = 315)	145 (46%)	170 (53.9%)		
61-80 (*n* = 425)	212 (49.8%)	213 (50.1%)			Total Cholesterol				
Height (cm)			159.9 ± 9.0	158.7 ± 13.3	High (*n* = 107)	60 (56%)	47 (43.9%)	4.37 ± 1.19	4.3 ± 1.1
Weight (kg)			80.2 ± 16.5	78.3 ± 17.1	Normal (*n* = 416)	220 (52.8%)	196 (47.1%)		
Duration of diabetes					Triglycerides				
1–10 years (*n* = 153)	72 (47%)	53 (34.6%)			High (*n* = 133)	82 (61.6%)	51 (38.3%)	1.62 ± 0.94	1.47 ± 0.65
11–20 years (*n* = 120)	87 (72.5%)	33 (27.5%)			Normal (*n* = 374)	198 (52.9%)	176 (47%)		
21–30 years (*n* = 78)	49 (62.8%)	29 (37.2%)							
>30 years (*n* = 34)	24 (70.5%)	10 (29.4%)			HDL	Median (IQR)			
BMI					High (*n* = 66)	44 (66.6%)	22 (33.3%)	1.19 (1.43-0.98)	1.13 (1.35–0.95)
Normal weight (*n* = 79)	31 (39.2%)	48 (60.7%)	31.3 ± 5.9	30.8 ± 6.1	Normal (*n* = 270)	148 (54.8%)	122 (45.1%)		
Overweight (*n* = 192)	110 (57.2%)	82 (42.7%)			Low (*n* = 158)	81 (51.2%)	77 (48.7%)		
Obese (*n* = 310)	155 (50%)	155 (50%)			LDL				
Underweight (*n* = 1)	0 (0%)	1 (0.2%)			High (*n* = 62)	32 (51.6%)	30 (48%)	2.67 ± 1.1	3.05 ± 4.6
Vitamin D deficiency					Normal (*n* = 462)	249 (53.8%)	213 (46.1%)		
Yes (*n* = 363)	261 (71.9%)	102 (28%)			Creatinine				
No (*n* = 237)	39 (16.4%)	198 (83.5%)			High (*n* = 69)	39 (56.5%)	30 (43.4%)	89.4 ± 63.1	81.8 ± 39.3
Consumption of Vitamin D supplements					Normal (*n* = 522)	259 (49.6%)	263 (50.3%)		
Yes (*n* = 490)	272 (55.5%)	218 (44.4%)							
No (*n* = 110)	28 (25.4%)	82 (74.5%)			Microalbumin Urine				
HbA1C					High (*n* = 159)	90 (56.6%)	69 (43.3%)	172 ± 530	182 ± 1238.6
<5.7% (*n* = 101)	43 (42.5%)	58 (57.4%)	7.7 ± 1.6	7.6 ± 1.7	Normal (*n* = 326)	176 (53.9%)	150 (46%)		
5.7–6.5% (*n* = 24)	16 (66.7%)	8 (33.3%)							
>6.5% (*n* = 461)	238 (51.6%)	223 (48.3%)			Creatinine Urine				
					High (*n* = 8)	3 (37.5%)	5 (62.5%)	7678 ± 5148	8036 ± 6017
					Normal (*n* = 440)	252 (57.2%)	188 (42.7%)		

Abbreviations: BMI, body mass index; HbA1c, glycated hemoglobin; CRP, C-reactive protein; HDL, high-density lipoprotein; LDL, low-density lipoprotein. Significant differences between group 1 (neuropathy) and group 2 (no neuropathy).

**Table 3 cells-12-00198-t003:** Significant differences between group 1 (neuropathy) and group 2 (no neuropathy).

Variables	Neuropathy	No Neuropathy	*p*-Value *
	*n* = 300	*n* = 300	
Age (years), mean ± SD	63 ± 9	62 ± 13	0.198
Age > 65, *n* (%)	151 (50)	160 (53)	0.462
Male sex, *n* (%)	124 (41)	123 (41)	0.934
BMI, *n* (%)			0.006
BMI 25–29.9	110 (37)	82 (27)	
BMI > 30	155 (52)	155 (52)	
Laboratory Data			
Vitamin D level (ng/mL), median (IQR)	20 (14–25)	33 (20–42)	<0.001
Vitamin D 20–30 (ng/mL), *n* (%)	100 (44)	41 (19)	<0.001
Vitamin D < 20 (ng/mL), *n* (%)	117 (39)	54 (18)	<0.001
HbA1C (10^9^/L), median (IQR)	7.4 (7–9)	7.1 (7–8)	0.279
Total cholesterol (mmol/L), median (IQR)	4.2 (3–5)	4.1 (4–5)	0.534
Triglycerides (mmol/L), median (IQR)	1.3 (1–2)	1.4 (1–2)	0.534
HDL (mmol/L), median (IQR)	1.2 (1–1)	1.2 (1–1)	0.534
LDL (mmol/L), median (IQR)	2.4 (2–3)	2.5 (2–3)	0.534
Serum creatinine (umol/L), median (IQR)	75 (61–96)	76 (62–92)	0.102
Urine creatinine (umol/L), median (IQR)	6361 (4402–10243)	6009 (3522–10761)	0.574
Microalbuminuria (mg/L), median (IQR)	11.4 (4–61)	11.1 (4–35)	0.020

* *p*-values < 0.05 were considered statistically significant. The bold text highlights significant differences between the two groups. Abbreviations: DM, diabetes mellitus; BMI, body mass index; HbA1c, glycated hemoglobin; HDL, high-density lipoprotein; LDL, low-density lipoprotein; IQR, interquartile range; SD, standard deviation.

**Table 4 cells-12-00198-t004:** Multivariate regression analysis for diabetic neuropathy.

Variables	Adjusted OR (95%CI)	*p*-Value *
Age, years	0.99 (0.97–1.01)	0.256
Male	0.87 (0.59–1.29)	0.491
Vitamin D < 20 (ng/mL)	2.63 (1.70–4.06)	<0.001
Log10 Microalbuminuria	1.40 (1.08–1.79)	0.010
BMI > 25	1.93 (1.09–3.41)	0.024

BMI, body mass index; CI, confidence interval; OR, odds ratio. * *p*-value < 0.05 were considered statistically significant.

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
