# Peer review of "Investigating the Association between Diabetic Neuropathy and Vitamin D in Emirati Patients with Type 2 Diabetes Mellitus"

_cells, 2023, doi:10.3390/cells12010198_

Round 1

Reviewer 1 Report

This is an important study highlighting the high prevalence of a neglected long-term complication of diabetes, namely diabetic neuropathy, in the UAE.

In relation to overall prevalence a recent multi-centre study highlighted the much higher prevalence of DPN and painful DPN in the Gulf region and also the association between, obesity and physical activity with DPN, painful DPN and between vitamin D deficiency and foot ulceration (Ponirakis et al. Prevalence and risk factors for diabetic peripheral neuropathy, neuropathic pain and foot ulceration in the Arabian Gulf region. J Diabetes Investig. 2022 Sep;13(9):1551-1559).

This study adds to the evidence vitamin D deficiency may play a role in DPN.

1.     This population has been selected from secondary care, which will have more complicated patients, which needs to be acknowledged.

2.     Also, the patient records were not randomly selected as the investigators selected those with (n=300) and without (n=300) DPN.

3.     How was DPN diagnosed and what about severity?

4.     Was there any assessment of painful DPN?

5.     What about DFU?

6.     The introduction lacks focus, which should be on the role of vitamin D in diabetic neuropathy, not diabetes in general.

7.     The English grammar and sentence construction need to be improved. “The mean age of the patients was 62.85± 11.35 years, and majority of these patients had been diabetes for more than 10 years. This is not good English, it should be ‘patients had diabetes’.

8.     What is the majority, why not simply state the duration of diabetes?

9.     Most of these patients (76.8%) had a HbA1c level of > 6.5%”. Either 23.2% were very well controlled or had IGT or did not have diabetes.

10.  “Vitamin D level of 50% of the patients was < 30 ng/mL. This should be broken down into exactly the percentage who were insufficient (<30), deficient (<20) or markedly deficient (<10).

11.  CRP, creatinine, microalbuminuria, and lipid profile including total cholesterol, HDL, LDL, and triglycerides were normal for majority of the patient”. Again, this is not precise expression of the results.

12.  In the table various cut-offs are used e.g., High, normal etc.- what is the definition of high or low for each variable?

13.  The discussion contains data and results from the study. These should be in the results section. The discussion is to discuss the outcomes compared to other findings.

14.  The current literature shows that circulation vitamin D (25-hydroxyvitamin D) could be involved in development and progression of diabetic neuropathy.” Should be circulating not circulation.

15.  The outcomes of the present study have found that individuals with T2DM with neuropathy were having lower vitamin D level than diabetic without neuropathy. Should be “The present study has shown that individuals with diabetic neuropathy had lower vitamin D levels compared to those without diabetic neuropathy.

16.  Patients with neuropathy had a low vitamin D level with a mean vitamin D level of 21.37± 11.9 240 ng/ml, while it was 33.41 ±15.7ng/ml in diabetic patient without neuropathy with a significant difference (P=< 0.001). This sentence should be in the results section and should read as “Vitamin D was significantly lower in patients with (21.37± 11.9 ng/ml) compared to without (33.41 ±15.7 ng/ml) diabetic neuropathy (P=< 0.001).

Author Response

08th November 2022 

Dear Chief Editor,

We would like to thank the editors and the reviewers for reading our submitted article and thoroughly reviewing the manuscript.

Please find the attached edited manuscript (File name: revised manuscript final draft) with the changes that have been suggested by the reviewers). 

Please find below the responses to the comments of the Editors and Reviewers.

Title: Investigating the association between diabetic neuropathy and vitamin D in Emirati patients with Type 2 Diabetes Mellitus

Authors: Tahra Abdalla Al Ali, Alizeh Ashfaq, Narjes Saheb Sharif-Askari, Salah Abusnana and Bashair M. Mussa*

Name of Journal: Cells

Manuscript NO:  1969365

Regards,

Bashair

REVIEWER #1 EVALUATION

This is an important study highlighting the high prevalence of a neglected long-term complication of diabetes, namely diabetic neuropathy, in the UAE.

In relation to overall prevalence a recent multi-centre study highlighted the much higher prevalence of DPN and painful DPN in the Gulf region and also the association between, obesity and physical activity with DPN, painful DPN and between vitamin D deficiency and foot ulceration (Ponirakis et al. Prevalence and risk factors for diabetic peripheral neuropathy, neuropathic pain and foot ulceration in the Arabian Gulf region. J Diabetes Investig. 2022 Sep;13(9):1551-1559).

This study adds to the evidence vitamin D deficiency may play a role in DPN.

Comment #1

This population has been selected from secondary care, which will have more complicated patients, which needs to be acknowledged.

Response

We would like to thank the reviewer for this comment. The following statement has been added to the methods section “Given that this is a secondary care unit, significant number of patients have complications including diabetic neuropathy”

Comment #2

Also, the patient records were not randomly selected as the investigators selected those with (n=300) and without (n=300) DPN.

Response

We would like to thank the reviewer for this comment.  The records were randomly selected from over thousand records and then divided into patients with diabetic neuropathy and the other group without neuropathy.

Comment # 3

How was DPN diagnosed and what about severity?

Response

We would like to thank the reviewer for this comment. 

Here are the diagnosis criteria: (i) patients were assessed for distal symmetric polyneuropathy, (ii) the assessment included a careful history   and either temperature or pinprick sensation (small-fiber function) and vibration sensation using a 128-Hz tuning fork (large-fiber function) and (iii) patients had an annual 10-g monofilament testing to assess for feet at risk for ulceration and amputation. Finally, electrophysiological testing or referral to a neurologist was rarely needed for screening, except in situations where the clinical features are atypical, the diagnosis is unclear, or a different etiology is suspected. Atypical features included motor greater than sensory neuropathy, rapid onset, or asymmetrical presentation.

Severity was not considered in the current analysis; however, in the future studies we can included to give more in depth information about the prognosis of the diabetic neuropathy.

Comment # 4

Was there any assessment of painful DPN?  

Response

We would like to thank the reviewer for this comment.  Assessment of the painful DPN was done by the endocrinologists, the current study is based on the medical records and the details that have been documented in these records.

Comment # 5

What about DFU?

Response

We would like to thank the reviewer for this comment. Diabetic foot ulcers were not included in the current ulcer as the study was focusing on diabetic neuropathy and Vitamin D. However, we can include it in the future studies.

Comment # 6

The introduction lacks focus, which should be on the role of vitamin D in diabetic neuropathy, not diabetes in general. 

Response

we would like to thank the reviewer for this comment. The introduction has been updated.

Comment # 7

The English grammar and sentence construction need to be improved. “The mean age of the patients was 62.85± 11.35 years, and majority of these patients had been diabetes for more than 10 years”. This is not good English, it should be ‘patients had diabetes’.

Response

We would like to thank the reviewer for this comment. The sentence has been changed to “The mean age of the patients was 62.85± 11.35 years, and majority of these patients had diabetes for more than 10 years”. 

Comment # 8

What is the majority, why not simply state the duration of diabetes?

Response

We would like to thank the reviewer for this comment. The sentence has been rephrased to “and had diabetes for more than 10 years.”

Comment # 9

“Most of these patients (76.8%) had a HbA1c level of > 6.5%”. Either 23.2% were very well controlled or had IGT or did not have diabetes.

Response

We would like to thank the reviewer for this comment.  The 23.2% of these patients had controlled diabetes.

Comment # 10

Vitamin D level of 50% of the patients was < 30 ng/mL”. This should be broken down into exactly the percentage who were insufficient (<30), deficient (<20) or markedly deficient (<10). 22,103,94

Response

We would like to thank the reviewer for this comment. The following sentence has been added, “Out of 600, 94 patients had vitamin D level of <20 ng/mL, 103 had vitamin D <30 ng/mL and 22 were markedly deficient in vitamin D.”

Comment # 11

“CRP, creatinine, microalbuminuria, and lipid profile including total cholesterol, HDL, LDL, and triglycerides were normal for majority of the patient”. Again, this is not precise expression of the results.

Response

We would like to thank the reviewer for this comment. The following explanation has been added, “CRP level of 53% of the patients was normal, including 87% of these patients had normal creatinine levels. 54% participants had normal microalbuminuria levels, and lipid profile including total cholesterol, HDL, LDL, and triglycerides were normal for majority of the patients. 69% of the patients had normal total cholesterol levels, 62% had normal triglyceride levels.”

Comment # 12

In the table various cut-offs are used e.g., High, normal etc.- what is the definition of high or low for each variable?

Response

We would like to thank the reviewer for this comment. The cut offs for the lab values have been added. HbA1c (normal: 6.5%), C- reactive protein levels( reference range: 0.0-9.0 mg/L), lipid profile including total cholesterol (normal range: 0.0-5.2 mmol/L), triglycerides (normal range: 0.40-1.82 mmol/L), HDL (normal range: > 1.04- 1.55mmol/L) and LDL ( normal range: < 0.00- 4.00 mmol/L). Serum creatinine (normal range: 62-115 umol/L), Urine creatinine (normal range: 8400-22000 umol/L) and urine microalbumin (normal range: 3-30 mg/L) were also included.

Comment # 13

The discussion contains data and results from the study. These should be in the results section. The discussion is to discuss the outcomes compared to other findings.

Response

We would like to appreciate the reviewer for this comment. The results have been removed from the discussion part.

Comment # 14

“The current literature shows that circulation vitamin D (25-hydroxyvitamin D) could be involved in development and progression of diabetic neuropathy.” Should be circulating not circulation

Response

We would like to thank the reviewer for this comment. The sentence has been changed to “The current literature shows that circulating vitamin D (25-hydroxyvitamin D) could be involved in development and progression of diabetic neuropathy.”

Comment # 15

 “The outcomes of the present study have found that individuals with T2DM with neuropathy were having lower vitamin D level than diabetic without neuropathy”. Should be “The present study has shown that individuals with diabetic neuropathy had lower vitamin D levels compared to those without diabetic neuropathy”.

Response

We would like to thank the reviewer for this comment. The sentence has been changed to “The present study has shown that individuals with diabetic neuropathy had lower vitamin D levels compared to those without diabetic neuropathy”.

Comment # 16

“Patients with neuropathy had a low vitamin D level with a mean vitamin D level of 21.37± 11.9 240 ng/ml, while it was 33.41 ±15.7ng/ml in diabetic patient without neuropathy with a significant difference (P=< 0.001)”. This sentence should be in the results section and should read as “Vitamin D was significantly lower in patients with (21.37± 11.9 ng/ml) compared to without (33.41 ±15.7 ng/ml) diabetic neuropathy (P=< 0.001).”

Response

we would like to thank the reviewer for this comment. The sentence has been added to the results section accordingly.

Reviewer 2 Report

This study is focused on Vitamin D levels in diabetes and whether there are differences between T2DM with and without peripheral neuropathy with the UAE population. The main outcomes show that Vitamin D deficiency is more prevalent in T2DM with neuropathy even with high levels of supplementation. Secondary variables of BMI and microalbumin appear to have a relationship with neuropathy.  

Abstract: should be updated to clarify that the group with neuropathy had lower Vit D levels than those without. That is the main point of the study and is not clear from the abstract.

Introduction/methods:

Needs a major review of grammar and language.

Introduction length could be reduced. 

Counterintuitive that UAE has limited sunlight hour exposure. May wish to qualify this statement.

Methods should report the threshold for deficient and non-deficient Vit D levels. Statements provided in the introduction, but helpful to clarify in the methods.

Methods should define normal levels of blood parameters (CRP, cholesterol levels)

Methods should include information on how the clinical diagnosis of diabetic neuropathy is made in the UAE.  

Vit D supplement dosage and type of supplement (if known) should be specified. Even it not asked, this should be made clear (was this a question of do you take or are you prescribed vitamin D supplements).

Unclear why enter regression was done instead of stepwise.

Results/discussion

Table layout needs updating.

Add symbol when group differences are significant (Vit D, HDL?).

Check SD for HDL (0.40 for neuropathy group and 9.3 for non-neuropathy group).

Not sure what the following statement means (line 188): 87% of the patients with neuropathy had a higher BMI, p=0.006 as shown in Table 3.” Is it correct that mean BMI values are not different between groups?

Table 3, HbA1C acronym needs updating

Not sure how Figure 1 adds to the results. Legend also needs updating.  

Normal CRP suggests lack of systemic inflammation, how can this be explained with low vitamin D levels especially with the neuropathy group.   

DNP (define acronym and should this be DPN?).

VDR (define) and provide better context.

Discussion should be more concise and the listing of others findings needs better comparison and context to compare with current results.

“He” needs updating (authors, plural)

Sever should be severe

Suggest to comment on the attention or frequency of Vit D testing in UAE.

Limitations include quantification of Vit D supplements. Generally, supplementation raises serum 25 (OH)D. Is it that supplementation type or dosage is too low or is there poor adherence?

Author Response

08th November 2022

Dear Chief Editor,

We would like to thank the editors and the reviewers for reading our submitted article and thoroughly reviewing the manuscript.

Please find the attached edited manuscript (File name: revised manuscript final draft) with the changes that have been suggested by the reviewers). 

Please find below the responses to the comments of the Editors and Reviewers.

Title: Investigating the association between diabetic neuropathy and vitamin D in Emirati patients with Type 2 Diabetes Mellitus

Authors: Tahra Abdalla Al Ali, Alizeh Ashfaq, Narjes Saheb Sharif-Askari, Salah Abusnana and Bashair M. Mussa*

Name of Journal: Cells

Manuscript NO:  1969365

Regards,

Bashair

REVIEWER #2 EVALUATION

This study is focused on Vitamin D levels in diabetes and whether there are differences between T2DM with and without peripheral neuropathy with the UAE population. The main outcomes show that Vitamin D deficiency is more prevalent in T2DM with neuropathy even with high levels of supplementation. Secondary variables of BMI and microalbumin appear to have a relationship with neuropathy. 

Comment # 1

Abstract: should be updated to clarify that the group with neuropathy had lower Vit D levels than those without. That is the main point of the study and is not clear from the abstract.

Response

We thank the reviewer for this comment. The Abstract has been changed accordingly.

Comment # 2

Introduction/methods:Needs a major review of grammar and language.

Response

We thank the reviewer for the comment. The grammar and language have been reviewed throughout the manuscript.

Comment # 3

Introduction length could be reduced.

Response

We thank the reviewer for this comment. The Introduction has been updated.

Comment # 4

Counterintuitive that UAE has limited sunlight hour exposure. May wish to qualify this statement.

Response

We appreciate the reviewer for this comment. The UAE has high sunlight exposure which may be a cause of vitamin D deficiency.

Comment # 5

Methods should define normal levels of blood parameters (CRP, cholesterol levels) Response

We would like to thank the reviewer for this comment. The cut offs for the lab values have been added. HbA1c (normal: 6.5%), C- reactive protein levels (reference range: 0.0-9.0 mg/L), lipid profile including total cholesterol (normal range: 0.0-5.2 mmol/L), triglycerides (normal range: 0.40-1.82 mmol/L), HDL (normal range: > 1.04- 1.55mmol/L) and LDL (normal range: < 0.00- 4.00 mmol/L). Serum creatinine (normal range: 62-115 umol/L), Urine creatinine (normal range: 8400-22000 umol/L) and urine microalbumin (normal range: 3-30 mg/L) were also included.

Comment # 6

Methods should include information on how the clinical diagnosis of diabetic neuropathy is made in the UAE. 

Response

We would like to thank the reviewer for this comment.  It is done and the following statements were added to the methods “(i) patients were assessed for distal symmetric polyneuropathy, (ii) the assessment included a careful history   and either temperature or pinprick sensation (small-fiber function) and vibration sensation using a 128-Hz tuning fork (large-fiber function) and (iii) patients had an annual 10-g monofilament testing to assess for feet at risk for ulceration and amputation. Finally, electrophysiological testing or referral to a neurologist was rarely needed for screening, except in situations where the clinical features are atypical, the diagnosis is unclear, or a different etiology is suspected. Atypical features included motor greater than sensory neuropathy, rapid onset, or asymmetrical presentation

Comment # 7

Vit D supplement dosage and type of supplement (if known) should be specified. Even it not asked, this should be made clear (was this a question of do you take or are you prescribed vitamin D supplements).

Response

We would like to thank the reviewer for this comment. The following sentence has been added, “Vitamin D supplementation which included Oligocalciferol, Ergocalciferol, Calciferol and Vitamin D3 were considered.”

Comment # 8

Unclear why enter regression was done instead of stepwise.

Response

We would like to thank the reviewer for this comment and apologize for the lack of clarity. For the variable selection, stepwise enter method was used. We corrected the method section.  

Comment # 9

Results/discussion: Table layout needs updating.

Response

We would like to thank the reviewer for this comment. The table layout has been updated according to the cells template.

Comment # 10

Add symbol when group differences are significant (Vit D, HDL?).

Response

We warmly thank the reviewer for raising this point. We have added * symbol for the significant values.  

Comment # 11

Check SD for HDL (0.40 for neuropathy group and 9.3 for non-neuropathy group).

Response

We appreciate the reviewer for this comment. As there was a huge difference in standard deviation, we have reported the median with IQR. 1.19 (1.43-0.98) for neuropathy group and 1.13 (1.35-0.95) for no neuropathy group.

Comment # 12

Not sure what the following statement means (line 188): 87% of the patients with neuropathy had a higher BMI, p=0.006 as shown in Table 3.” Is it correct that mean BMI values are not different between groups?

Response

We would like to thank the reviewer for this comment. The BMI values were approximately same in both groups.

Comment # 13

Table 3, HbA1C acronym needs updating

Response

We would like to thank the reviewer for this comment. The HbA1c acronym has been updated.

Comment # 14

Not sure how Figure 1 adds to the results. Legend also needs updating.  

Response

We thank the reviewer for this comment. Legend was updated and the Pearson correlation results were added to the figure. Figure 1 was used to illustrate graphically the relationship between the low Vitamin D level and diabetic complications of microalbuminuria as well as diabetic neuropathy.  

Comment # 15

Normal CRP suggests lack of systemic inflammation, how can this be explained with low vitamin D levels especially with the neuropathy group. 

Response

We thank the reviewer for this comment.  CRP was high in ~ 60% of patients with diabetic neuropathy. Previous studies have shown that the relationship between increased CRP and diabetic neuropathy was clearer in specific types as painful neuropathy (Ref. 1). 

References

  1. Doupis J, Lyons TE, Wu S, Gnardellis C, Dinh T, Veves A. Microvascular reactivity and inflammatory cytokines in painful and painless peripheral diabetic neuropathy. J Clin Endocrinol Metab. 2009 Jun;94(6):2157-63. doi: 10.1210/jc.2008-2385

Comment # 16

DNP (define acronym and should this be DPN?).

Response

We would like to thank the reviewer for this comment. There was a mistake, the acronym is DPN for diabetic polyneuropathy.

Comment # 17

VDR (define) and provide better context. 

Response

We thank the reviewer for this comment. The VDR definition and context has been updated in the manuscript.

Comment # 18

Discussion should be more concise and the listing of others findings needs better comparison and context to compare with current results.

Response

We appreciate the reviewer for this comment. The discussion has been updated.

Comment # 19

“He” needs updating (authors, plural)

Response

We would like to thank the reviewer for this comment. The ‘He’ has been changed to ‘They’.

Comment # 20

Sever should be severe

Response

We would like to thank the reviewer for this comment. The word has been changed to severe.

Comment # 21

Suggest to comment on the attention or frequency of Vit D testing in UAE.

Response

We thank the reviewer for this comment. Vit D is checked frequently in UAE as there is high sun exposure and most of the population is deficient in Vitamin D.

Comment # 22

Limitations include quantification of Vit D supplements. Generally, supplementation raises serum 25 (OH)D. Is it that supplementation type or dosage is too low or is there poor adherence?

Response

We would like the thank the reviewer for this comment. We think that there is lack of adherence to vitamin D supplements among the UAE population.

Round 2

Reviewer 1 Report

Thank you for addressing my concerns.